# Hematological parameters in a population of male bakers exposed to high heat work environment

**Sultan T. Al-Otaibi** 📷 *

Associate Professor of Occupational Medicine, Department of Public Health, College of Public Health, Imam Abdulrahman Bin Faisal University, Dammam, Saudi Arabia

* salotaibi@iau.edu.sa

## Abstract

### Objectives

The aim of this study was to determine the hematological changes associated with heat exposure in a population of bakers.

### Materials and methods

Personal information was collected using a questionnaire, and a venous blood sample was drawn at the end of a work shift from the bakers and from a control group. The average wet-bulb globe temperature (WBGT) index was measured in the workplaces of both the bakers and the controls.

### Results

This cross-sectional study involved 137 bakers working in 20 bakeries and 107 controls who were comparable in terms of likely confounding factors. Hemoglobin and platelet values were abnormal among the bakers and statistically significantly different to the control group (P = 0.026, and P = 0.016 respectively). The average WBGT index in the bakeries was 37.4˚C, while the average WBGT in the workplaces of the controls was 25.5˚C, (P<0.0001).

### Conclusions

The changes in the bakers' hematological parameters were found to be associated with exposure to high environmental heat at bakeries, as measured by the WBGT index. Preventive measures should be introduced to reduce the adverse effect of heat exposure among bakers and directed toward the worker–equipment–environment triad.

**Data Availability Statement:** Due to confidentiality restrictions imposed by the Eastern Region Municipality of Saudi Arabia, the data underlying this study may not be made publicly available. However, the data may be accessed upon request

to info@eamana.gov.sa by interested and qualified researchers who meet the criteria for access to confidential data. The authors did not have any special privileges in accessing the data that other researchers would not have.

**Funding:** The authors received no specific funding for this work.

**Competing interests:** The authors have declared that no competing interests exist.

## Introduction

Exposure to high heat has a determining impact on the strength of unprotected people, especially those working in occupations characterized by high environmental temperatures, and this has ramifications for developing nations in which skilled workers may be less common and their replacement therefore troublesome [1,2]. Heat disorders include heat stroke, heat exhaustion, heat syncope, heat cramps, and behavioral disorders, as well as injuries [2–8].

Bakers are additionally at high risk of developing musculoskeletal illnesses from repetitive movements during their daily work, which involves handling dough in awkward postures, heavy lifting, and long shift-working hours [9–12].

Employees exposed to high heat were at risk of developing physiological changes (increased heart rate, respiratory rate, core body temperature, and blood pressure), which can be explained by sweating resulting in dehydration [13]. Furthermore, male infertility among workers were reported to be associated with heat exposure [14].

Previous studies have reported that heat stress increases reactive oxygen species and induces high levels of malondialdehyde (MDA) and nitric oxide (NO), causing hematological changes (abnormal mean cell volumes [MCV], white blood cells [WBC], red blood cells [RBC], and hemoglobin [Hb]) among exposed workers. Increases in body temperature also raise the level of NO synthase, subsequently releasing NO [15–17].

There are several heat stress indexes used to assess exposure to high environmental temperatures [18,19], and the wet-bulb globe temperature (WBGT) has been validated and is used globally as an acceptable measure of environmental heat [20,21].

Although there is a significant body of research on heat exposure in the workplace, there is limited information on hematological responses associated with exposure to high environmental temperatures among bakers, which this study therefore sought to examine.

The hypothesis of this study was to find a statistical difference in hematological parameters among bakers exposed to heat compared to a comparison group.

The rationale of this study: there was a dearth of research on the effect of heat exposure at the workplace in general, and in particular, there were few studies on hematological changes among workers exposed to high environmental heat. It is hoped that this study will contribute to the medical literature.

On the other hand, the objective of this study was to determine the hematological changes associated with heat exposure in a population of bakers and compare it to control group.

## Methods

This was a cross-sectional study of all 137 bakery workers from 20 bakeries in the city of Al-Khobar, Saudi Arabia. Eleven of eligible bakers were either not present at the time of the study and or they were on vacation when this study conducted giving a response rate of 93 percent.

**Inclusion criteria:** for the bakers were work experience in a bakery of at least one year, had no history of medical conditions and currently were not taken medications.

**Exclusion criteria:** bakers working less than one year, had history of cardiovascular diseases, and using current medications, and or had history of medical conditions that may affect blood hemogram parameters.

The comparison group of 107 participants was selected from individuals who had the following:

a. No exposure to heat in their current jobs; the participants were salesmen in offices, butchers, janitors, and others.

b. No present or past history of work demanding exposure to high environmental temperatures.

c. A pattern of working long hours, including night and early morning shifts, to ensure comparability in this regard with bakers. Most of the control group subjects were selected because they performed physical work with a similar energy expenditure to the bakers.

All participants were invited to complete a questionnaire in English (S1 Appendix) that assessed personal information (age, nationality, marital status, educational level, smoking, and income), work history, job title, work shift, working hours and duration of works in years), medical history, medication taken, fluids consumed during a work shift, and type of clothing.

Bakers who did not speak English or those illiterates were interviewed through interpreter.

A venous blood sample was drawn from the antecubital fossa of all participants at the end of their work shifts. All samples were sent immediately to a central laboratory, where they were processed within one hour using an automated hematology analyzer (Coulter Counter Model 5800; Beckman Coulter Inc, USA) to measure the hematological parameters. It had originally been planned to draw blood samples from all workers both before and after the work shift, but this proved unacceptable to the workers during the pilot study.

Measurements of the WBGT index, which is measured in degrees Celsius, were performed according to ISO7243 standards using a WBGT meter (8778 AZ; AZ Instrument Corp, Taiwan) and were taken simultaneously at the different locations at which the workers performed their routine job duties. The relative humidity was read off the psychometric chart. The average relative humidity in bakeries was 27%±15%. The study was conducted during the summer, when the average local outside air temperature is 33˚C. The data were checked and recorded daily, and the Statistical Package for Social Sciences (SPSS) version 20 was used for the analysis. T-tests were used to compare the means of the hematological parameters between the bakers and the control group, and chi-squared ($\chi^2$) tests and Fisher's exact test were used to compare differences in the percentages of categorical variables. ANOVAs and both univariate and multiple logistic regression analyses were also conducted, and Tukey's test was used as a post hoc. P-values of less than 0.05 were taken to be statistically significant.

## Ethical considerations

Imam Abdulrahman bin Faisal University's ethical review board (IRB-2018-03-194) approved the study. The management of all bakeries in the study area were then contacted, and their cooperation requested. Written consent was obtained from all participants after explaining the aims of the study. Participants were told they were free to withdraw from the study at any time, and they were assured of the confidentiality of the recorded data. All procedures involving human participants were conducted in accordance with the ethical standards of the relevant institutional and national research committees and with the 1964 Helsinki Declaration and its later amendments or comparable ethical standards.

## Results

All bakers in this study worked for more than one year in the bakeries and were healthy and had no history of medical conditions that may affect blood hemogram parameters.

The 137 bakers were compared to the 107 controls in terms of age, gender (All bakers and controls were males), marital status, nationality, education, duration of work, smoking habits, and income:

- Age: The mean age of the bakers was 32.3 years, with a standard deviation (SD) of 7 years, and the mean age for the controls was 30.5 years, with a SD of 7.5 years. (P = 0.054). Age

group was distributed as follow: those of age 23–26 years were 32% among bakers and 31% among the control group while those of age 27–30 years were 29% bakers and 31% of the control group. On the other hand, those above the age of 30 years were 39% among the bakers and 37% among the control group.

- Marital status: 81.0% of the bakers and 72.9% of the controls were married; the remainder were single (19% and 27.1%), (P = 0.132).

- Nationality: 67.8% of the bakers were Indian subcontinent (India, Pakistan, and Bangladesh), 11.7% were Filipino, and 20.5% were Arabs On the other hand, 56.1% of the control group were Indian subcontinent, 14.0% were Filipino, and 29.9% were Arabs (P = 0.221).

- Level of education: Of the bakers, 22.2% were illiterate, 30.7% had a primary school education, and 19.7% had a secondary school or higher education; 22.4% of the controls had a secondary school or higher education (P = 0.644).

- Job experience: The mean duration of the bakers' work was 4.7 (SD: 2.5) years, compared to 4.0 (SD: 2.8) years for the controls (P = 0.534). In this study, 57% of the bakers were baking different types of bread, 34% of them were dough mixers and the rest of them (9%) were decorating cake with toppings.

- Smoking: Of the bakers, 63.5% were current smokers, compared to 52.3% of the controls; 9.5% of the bakers were ex-smokers (stopped smoking more than a month before the study), compared to 14.0% of the controls; and the remainder had never smoked (P = 0.198). The mean number of cigarettes smoked per day among the bakers was 9.9 (SD: 10.9), compared to 10.8 (SD: 12.7) among the controls (P = 0.544).

- Income: The average income of the bakers was US dollars 295 per month compared to US dollars 357 for the controls (P = 0.076).

- The diagnosis of obesity in our study was obtained by the Body Mass Index (BMI), and was calculated as follows: BMI = weight in kg/(height in meters)$^2$. Using the BMI method, the mean BMI for overweight and obese subjects was found to be 29.40 at the end of the work shift among bakers compared to 30.11 among the control group (P = 0.341).

Therefore, with no statistically significant differences, the two groups (bakers and control) were considered comparable in terms of the assessed confounding factors.

All bakers in this study worked for more than one year in the bakeries and were healthy and had no history of medical conditions that may affect blood hemogram parameters.

Table 1 shows that the bakers' average Hb (9.92 MMOL/L) was statistically significantly higher than that of the controls (9.75 MMOL/L; P = 0.026). The bakers' mean platelet levels (243.04×10$^9$/L) were also slightly, but significantly, lower than those of the controls

**Table 1. Hematological parameters among the bakers and the control group.**

| Parameter | Bakers | | Control group | | P-value |
|---|---|---|---|---|---|
| | Mean | S.D | Mean | S.D | |
| Hb (MMOL/L) | 9.92 | 0.65 | 9.75 | 0.69 | 0.026 |
| RBC (X10$^{12}$/L) | 5.22 | 0.47 | 5.24 | 0.46 | 0.8.19 |
| WBC (X10$^9$/L) | 7.95 | 2.05 | 8.06 | 2.17 | 0.689 |
| Platelets (X10$^9$/L) | 243.04 | 53.75 | 258.32 | 57.34 | 0.016 |

Hemoglobin (Hb), Red Blood Cell (RBC), White Blood Cell (WBC), Standard Deviation (S.D).

**Table 2. Comparison of hemoglobin levels among the bakers and the control group.**

| Parameter | Bakers | | Control group | | P-value |
|---|---|---|---|---|---|
| | Number | % | Number | % | |
| Low (<8.7 MMOL/L) | 2 | 1.5 | 6 | 5.6 | 0.025 |
| Normal (8.7–10.6 MMOL/L) | 116 | 84.6 | 91 | 85.0 | |
| High (>10.6 MMOL/L) | 19 | 13.9 | 10 | 9.4 | |
| Total | 137 | 100 | 107 | 100 | |

$(258.32 \times 10^9$/L; $P = 0.016$). There was no statistically significant difference between the means of the RBC and WBC levels. Table 2 shows that 13.9 percent of the bakers had an Hb level higher than 10.6 MMOL/L, compared to 9.4 percent among the controls, and this intergroup difference was significant ($P = 0.025$).

A significant difference in WBC between nationalities was found using ANOVA which yielded a variance ratio F = 8.9467 ($P = 0.0002$). Further analysis by multiple post hoc comparisons using Tukey's test as shown in Table 3, indicated a significant difference between the Indian subcontinent and other nationalities ($P < 0.05$). There was no correlation between increase hemoglobin label and decrease mean platelet volume. On the other hand, no other relationships between the measured hematological parameters and the personal characteristics of the studied population.

Of the bakers, 79.4 percent drank less fluids during work, compared to 90.5 percent of the controls ($P = 0.014$). The frequency of fluid-drinking among the bakers was lowest (9.4 percent) 30 minutes into the workday and gradually increased, but still with few taking fluids (37.6 percent) 90 minutes after starting work. The controls showed the reverse pattern, with 25 percent at 30 minutes and 11.3 percent at 90 minutes ($P < 0.001$).

The bakers were exposed to heat for 10 hours each day, 7 days a week, with a meal break of 30 minutes. They also worked long hours, including night and early morning shifts, and under time pressure such that they could not rest nor drink during work.

The bakers had a mean total clothing insulation (Total $I_{cl}$ = 1.23 clo) lower than the controls (1.32 clo; $P < 0.0001$), meaning that the bakers wore lighter clothes. The mean WBGT in the bakeries was 37.4°C, compared to 25.5°C in the workplaces of the controls ($P < 0.0001$).

## Discussion

In this study, hematological changes and environmental conditions among bakery workers in the city of Al-Khobar of Saudi Arabia were monitored. Heat stress levels were measured using the WBGT to evaluate the impact of environmental factors on the bakers and a control group. Bakers in this study were acclimatized to heat by virtue of their work exposure, which by definition the controls were not. However, it has been reported in previous studies that heat acclimatization does not protect even trained individuals from the impact of exposure to environmental heat [22,23].

**Table 3. Post hoc test for mean score of WBC by nationality among the participants.**

| Group1 | Group 2 | P value |
|---|---|---|
| Indian Subcontinent | Filipino | 0.014 |
| Indian Subcontinent | Arabs | 0.012 |
| Arabs | Filipino | 0.243 |

Previous studies reported that there were changes in the hematological parameters (Hb, RBC, WBC, and platelets) of workers exposed to high environmental temperatures compared to the control group [24,25].

The average Hb level among bakers was statistically significantly higher than the control group, which was in line with other studies which can be explained by sweating resulting in dehydration [17,26]. Our study indicated that RBC and WBC levels were lower among the bakers than the controls, but this was not statistically significant, which is consistent with a previous study among workers exposed to high environmental heat [26]. Similar reductions in RBC and WBC have been reported in animal studies after exposure to high heat, and this was explained as being caused by tissue damage that was dependent on the level and duration of the heat exposure [27,28]. It should be noted that animal studies might be problematic to explain this issue, because humans are not fish and they usually do not have fur. If Hb increases and RBC stays almost the same: Might it be, that not only the plasma volume is shrinking but the volume of the erythrocytes too? Is it really tissue damage or might it be, that this is a physiological response to survive extreme conditions? On the other hand, the reduction of platelets might be a protective mechanism to circumvent higher blood viscosity and thromboembolic events.

The mean platelet levels in this study were significantly lower among bakers than the control group, and similar findings have previously been reported among bakers exposed to high environmental temperatures [17]. Bakers in this study had hematological changes that can be explained by exposure to heat, and this was in line with the results of previous studies [17,26]. The bakers in this study who reported low levels of fluid consumption at work were also more likely to experience heat-related hematological changes, which is consistent with previous research [17].

The bakeries in this study exceeded the American Conference of Governmental Industrial Hygienists (ACGIH) threshold limit value for working in hot environments, as measured by the WBGT index [29], indicating that the bakers work under high environmental temperatures, which is consistent with other studies in which a significant relationship was found between WBGT and hematological parameters [17,26]. This investigation found that heat exposure impacts the wellbeing of bakery specialists in Saudi bread kitchens. Improvements in working conditions and laborer training are key changes needed to avoid harming bread cooks' health.

## Conclusions

Hematological parameters were abnormal (high hemoglobin and low platelet values) among the bakers who were exposed to heat in this study, and these levels were found to be statistically significantly different to a control group. High environmental temperatures were recorded in Saudi bakeries, as measured by the WBGT index, which would explain the findings of this study. Preventive measures should be aimed at reducing the adverse effects of heat exposure among bakers and should be directed toward the worker–equipment–environment triad.

It is hoped that this study will contribute to the medical literature since there were only very few studies published in this regard.

### Limitations of the study

The strength of our study assured that data were collected all at once, it's less likely that participants will quit the study before data were fully collected and this would increase the validity of our findings.

As with other cross-sectional studies, this study was susceptible to survivor bias because it assessed prevalent, rather than incident, cases and thus did not consider those who had retired or resigned. Furthermore, this study was subjected to potential biases, such as recruitment, non-response, exposure assessment, and statistical analysis. A control group twice the size of the experimental group was originally envisaged, but substantial difficulties were encountered and so a one-to-one ratio was used instead.

The personal information may have been subject to reporting bias, as it was collected by means of a self-report questionnaire. Blood samples were drawn from all participants while at work, instead of before and after, so we were unable to examine changes over the course of the day in the hematological parameters of the bakers. This study also involved only male participants, so no gender difference analysis was possible; however, this does not represent a sampling bias as it accurately reflects the male-only nature of this occupation in Saudi Arabia.

## Supporting information

**S1 Appendix.**
(DOCX)

## Acknowledgments

The author would like to thank all bakers and control groups participated in this study.

## Author Contributions

**Conceptualization:** Sultan T. Al-Otaibi.

**Data curation:** Sultan T. Al-Otaibi.

**Formal analysis:** Sultan T. Al-Otaibi.

**Funding acquisition:** Sultan T. Al-Otaibi.

**Investigation:** Sultan T. Al-Otaibi.

**Methodology:** Sultan T. Al-Otaibi.

**Project administration:** Sultan T. Al-Otaibi.

**Resources:** Sultan T. Al-Otaibi.

**Software:** Sultan T. Al-Otaibi.

**Supervision:** Sultan T. Al-Otaibi.

**Validation:** Sultan T. Al-Otaibi.

**Visualization:** Sultan T. Al-Otaibi.

**Writing – original draft:** Sultan T. Al-Otaibi.

**Writing – review & editing:** Sultan T. Al-Otaibi.

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
