## [Decision Letter · Decision Letter 0]

5 Jul 2022

PONE-D-22-06587Hematological Parameters in a Population of Male Bakers Exposed to High Heat Work Environment PLOS ONE

Dear Dr. Al-Otaibi,

Thank you for submitting your manuscript to PLOS ONE. After careful consideration, we feel that it has merit but does not fully meet PLOS ONE’s publication criteria as it currently stands. Therefore, we invite you to submit a revised version of the manuscript that addresses the points raised during the review process. In particular, reviewer 1's comment about other conditions influencing hemotalogical parameters and their consideration in study design and analysis need to be addresssed. The discussion needs to reflect strengths and limitations of your study as well as a constructive discussion against the literature instead of just repeating results. Please also address the reviewer comments about the applicability of an English questionnaire among workers of whom 22% were illiterate. Please explain your calculation of the response proportion (elgible subjects, contacted subjects, participants),  because 100% response among elegible subjects appears highly unlikely.

Please also check the p-value for the comparison of smokers (p=0.198), which appears questionable, given that 63.5% of bakers were smokers compared to 52% of controls.

After addressing all comments, please submit your revised manuscript by Aug 19 2022 11:59PM. If you will need more time than this to complete your revisions, please reply to this message or contact the journal office at plosone@plos.org. Please include the following items when submitting your revised manuscript:A rebuttal letter that responds to each point raised by the academic editor and reviewer(s). You should upload this letter as a separate file labeled 'Response to Reviewers'.A marked-up copy of your manuscript that highlights changes made to the original version. You should upload this as a separate file labeled 'Revised Manuscript with Track Changes'.An unmarked version of your revised paper without tracked changes. You should upload this as a separate file labeled 'Manuscript'.

We look forward to receiving your revised manuscript.

Kind regards,

Thomas Behrens

Academic Editor

PLOS ONE

Journal Requirements:

"None declared"

Reviewers' comments:

Reviewer's Responses to Questions

**Comments to the Author**

1. Is the manuscript technically sound, and do the data support the conclusions?

Reviewer #1: No

Reviewer #2: Partly

Reviewer #3: Yes

2. Has the statistical analysis been performed appropriately and rigorously? 

Reviewer #1: No

Reviewer #2: No

Reviewer #3: Yes

3. Have the authors made all data underlying the findings in their manuscript fully available?

Reviewer #1: Yes

Reviewer #2: No

Reviewer #3: Yes

4. Is the manuscript presented in an intelligible fashion and written in standard English?

Reviewer #1: No

Reviewer #2: Yes

Reviewer #3: Yes

5. Review Comments to the Author

Reviewer #1: The manuscript, PONE-D-22-06587, is reviewed. There are many flaws with the paper that must be raised.

1. Statements in result of the abstract needs a p value.

2. Rationale of the study is not properly mentioned.

3. Design of the study is the greatest flaw. Exclusion criteria is not clear. Many conditions may effect blood hemogram parameters. Were they excluded or not?

4. Statistics were not expressed properly. For instance, it is unclear from the text which normality test was applied to the study variables.

5. Presentation of results is in a form that is difficult to understand.

6. Discussion is just a repetition of the results rather than being a constructive discussion of them along with literature data.

7. Inappropriate self-citations were noted.

Reviewer #2: Dear Sir,

thank you very much for your very interesting research. I encourage you to read the following comments as suggestions to improve your publication. They are not meant to discourage you. Due to cultural differences between Saudi-Arabia and Germany, it might be possible that my direct way to point out things could feel offensive to you. Be assured they are not. I appreciate your work very much.

Please take into account that your tables should not be mandatory for understanding, because the text should reflect all major findings and the core of your data.

I understood that the participating bakers had significantly higher Hb-levels and lower platelet-counts than the control-group. These findings are associated with significantly higher WGB-temperatures of the bakers work environment and lower rehydration than in the control group. These are your main findings as far as I understood. But with in the "Results"-section these findings are hidden behind other data like marital status, nationality and level of education. Why? Wouldn't it be better to summarise age and BMI in a table with the statement that groups are comparable and then to present your central findings? I think your work is about extreme physical influences like temperature and humidity at the human body. But it is not a sociological description study. In my view you should focus at data which are relevant to your hypothesis.

I do not understand the benefits or scientific value of a comparison of e.g. marital status, nationality, income and level of education. The level of illiterates off 22.4% could devalue your work because readers could interpret this as more than a fifth of your bakers were not able to fill in your questionnaire in their own language. Not to speak of English language.

By the way: I don't think it is appropriate to have sentences like "It is believed that (...)" with a reference in the "Results"-section. This belongs to the "Discussion"-section.

Further I do not understand what was your hypothesis? Had there been a power calculation with the pilot study you mentioned or other scientific work?

Please take into account that usually it is not appropriate to collect data and then test "wildly" for statistical differences. You should articulate a hypothesis generated with preceding findings of yours or others and then test for it.

In the "discussion"-section you write: "...reductions in RBC and WBC have been reported in animal studies after exposure to high heat, and this was explained as being caused by tissue damage that was dependent on the level and duration of the heat exposure." But your data does not support this! And animal studies might be problematic in this particular topic, because humans are not fish and they usually do not have fur.

It might be useful to concentrate at the Hb instead: If Hb increases and RBC stays almost the same: Might it be, that not only the plasma volume is shrinking but the volume of the erythrocytes too? Is it really tissue damage or might it be, that this is a physiological response to survive extreme conditions? E.g., might the reduction of platelets be a protective mechanism to circumvent higher blood viscosity and thromboembolic events?

Could you provide data of the amount of fluids the participants were drinking? I think this might be more relevant than the time/frequency they were drinking.

Could you make a subgroup analysis with bakers with the hottest working conditions compared with the less hot working conditions? And the same with high and low fluid intake during the shift, respectively?

I couldn't develop any feeling about the real work environment of the bakers: Was it hot & dry? Or more humid? The WBGT is not telling that. Could you provide additional data about humidity, because this significantly alters the ability of the human body to adopt to high temperatures?

At scientific papers like PLOS usually it is not appropriate to cite whole medical textbooks. If there is a scientific finding you cannot find anywhere else (primary scientific paper), you should at least provide the exact page(s) where the fact could be found.

Than there is another point in your data which is not self-explanatory: Why the bakers are so young and why is their working experience so low? Do they drop out their jobs because of the high physical demands or because they earn enough money to go back to their home countries or do they get promoted to higher positions in their bakeries without having these working conditions?

Thank you again for your interesting paper. I hope my comments can help you. Keep up the good work!

Reviewer #3: This is an article that studies the hematological pattern in bakers exposed to heat stress, compared with workers who are not under this type of occupational risk condition. The objective was achieved using adequate methodology. However, the manuscript is insufficient to presenting results and discuss it properly.

Regarding the statistically significant difference in hemoglobin and platelet values, the authors need to explain how such findings are compatible with the literature on exposure to heat stress, since terms such as "sweting resulting in dehydration" or "tissue damage" are very unspecific to explain the physiological process. Mainly considering that there was no difference on white or red cells. And Why hematocrit data was not presented, which is directly related to dehydration?

I suggest to include a table with the analysis about data collected, specially regarding work characteristics - such as working hours and exposure time, which are crucial to understanding the impact of occupational hazards. The authors chose to present only the significant difference in the nationality of the participants, but they do not present a discussion for this finding - was it a random result?

Finally, the study could present more information about exposure to heat stress evaluation, as there is only the report of the average among exposed and controls. Maybe the details of the range of temperature variation can be more explored in discussion about excessive exposure.

Reviewer #1: **Yes: **Gulali Aktas

Reviewer #2: No

Reviewer #3: No

---

## [Author Response · Author response to Decision Letter 0]

5 Aug 2022

Please see attached response to editor and reviewers.

---

## [Decision Letter · Decision Letter 1]

25 Aug 2022

PONE-D-22-06587R1Hematological Parameters in a Population of Male Bakers Exposed to High Heat Work EnvironmentPLOS ONE

Dear Dr. Al-Otaibi,

Thank you for submitting your manuscript to PLOS ONE. After careful consideration, we feel that it has merit but based on the recent reviewer comments, does not fully meet PLOS ONE’s publication criteria as it currently stands. Therefore, we invite you to submit a revised version of the manuscript that addresses the points raised during the review process. In particular, please refer again to Reviewer 1's original comments that need to be considered in more detail. As Reviewer 3 has noted, the limitations section should focus more on potential biases, such as recruitment, non-response, exposure assessment, and statistical analysis.

We look forward to receiving your revised manuscript.

Kind regards,

Thomas Behrens

Academic Editor

PLOS ONE

Journal Requirements:

Reviewers' comments:

Reviewer's Responses to Questions

**Comments to the Author**

1. If the authors have adequately addressed your comments raised in a previous round of review and you feel that this manuscript is now acceptable for publication, you may indicate that here to bypass the “Comments to the Author” section, enter your conflict of interest statement in the “Confidential to Editor” section, and submit your "Accept" recommendation.

Reviewer #1: (No Response)

Reviewer #3: All comments have been addressed

2. Is the manuscript technically sound, and do the data support the conclusions?

Reviewer #1: Partly

Reviewer #3: Yes

3. Has the statistical analysis been performed appropriately and rigorously? 

Reviewer #1: No

Reviewer #3: Yes

4. Have the authors made all data underlying the findings in their manuscript fully available?

Reviewer #1: Yes

Reviewer #3: Yes

5. Is the manuscript presented in an intelligible fashion and written in standard English?

Reviewer #1: Yes

Reviewer #3: Yes

6. Review Comments to the Author

Reviewer #1: Authors somewhat improved the quality of the paper in revised version. However significant proportion of my criticisms were not addressed adequately.

Reviewer #3: Authors answered questions sent in the #1 round. The reviewed manuscript brings more information about the research and go further in discussing the topic. I do not believe that the "objective" and "hypothesis" included (line 72-75) improved the paper. Maybe limitations session should be focused in biases, such as recrutation, losses, exposure, evaluation, analyze.

7. PLOS authors have the option to publish the peer review history of their article (what does this mean?). If published, this will include your full peer review and any attached files.

Reviewer #1: No

Reviewer #3: No

---

## [Author Response · Author response to Decision Letter 1]

1 Sep 2022

Please note attached response to reviewers

---

## [Editor Report · Decision Letter 2]

6 Sep 2022

Hematological Parameters in a Population of Male Bakers Exposed to High Heat Work Environment

PONE-D-22-06587R2

Dear Dr. Al-Otaibi,

We’re pleased to inform you that your manuscript has been judged scientifically suitable for publication and will be formally accepted for publication once it meets all outstanding technical requirements.

Kind regards,

Thomas Behrens

Academic Editor

PLOS ONE
---

## [Editor Report · Acceptance letter]

8 Sep 2022

PONE-D-22-06587R2 

Hematological Parameters in a Population of Male Bakers Exposed to High Heat Work Environment 

Dear Dr. Al-Otaibi:

I'm pleased to inform you that your manuscript has been deemed suitable for publication in PLOS ONE. Congratulations! Your manuscript is now with our production department. 

Kind regards, 

on behalf of

Prof. Thomas Behrens 

Academic Editor

PLOS ONE